**Subject Category:**
Biology (whole organism)

microbiology/health and disease and epidemiology

raw meat, pet food, pathogens, antimicrobial resistance, animal and public health

**Author for correspondence:**
Magdalena Nüesch-Inderbinen
e-mail: magdalena.nueesch-inderbinen@uzh.ch

# Raw meat-based diets for companion animals: a potential source of transmission of pathogenic and antimicrobial-resistant Enterobacteriaceae

Magdalena Nüesch-Inderbinen, Andrea Treier,
Katrin Zurfluh and Roger Stephan

Institute for Food Safety and Hygiene, Vetsuisse Faculty University of Zurich,
Winterthurerstrasse 272, 8057 Zurich, Switzerland

(iD) MN-I, 0000-0002-3242-9739; RS, 0000-0003-1002-4762

Feeding pets raw meat-based diets (RMBDs) has become increasingly popular but may constitute a risk due to the contamination with pathogenic and antimicrobial-resistant (AMR) bacteria. The aim of this study was to evaluate commercially available RMBDs with regard to microbiological quality and occurrence of AMR Enterobacteriaceae. Of 51 RMBD samples, 72.5% did not meet the microbiological standards for Enterobacteriaceae set out by EU regulations for animal by-products intended for pet food. Furthermore, *Salmonella* was detected in 3.9% of the samples. AMR bacteria were found in 62.7% of the samples, the majority thereof were resistant to third-generation cephalosporins due to the production of extended-spectrum β-lactamases (ESBLs) including CTX-M-1, which is widespread in livestock, and CTX-M-15, which is the most common ESBL variant worldwide. Colistin- and aminoglycoside-resistant isolates, producing MCR-1 and RMTB, were identified in 3.9 and 2% of the samples, respectively. The majority of the AMR *Escherichia coli* belonged to commensal groups A or B1 and were associated with clonal complexes CC155 and CC10. Two belonged to the emerging extraintestinal pathogenic CC648, and one to the globally disseminated uropathogenic *E. coli* sequence type ST69, suggesting zoonotic potential. The microbiological quality and the high prevalence of AMR producing Enterobacteriaceae in RMBDs raise concerns for animal and public health.

# 1. Introduction

Raw pet food has become increasingly popular among dog owners seeking to feed their pets on what is perceived as a natural and healthy diet. Health claims include benefits to canine vitality, the digestive tract and the immune system, but currently lack scientific evaluation [1]. Raw meat-based diets (RMBDs), also known as biologically appropriate raw food (BARF), consist mainly of raw muscle meats, organ meats and meaty bones. Some diets additionally contain vegetables, fruits or grain. Like conventional pet food, most RMBDs are based on the by-products of animals slaughtered for human consumption; however, they do not contain additives and supplements such as preservatives, stabilizers, gelling agents, sweeteners, flavours or vitamins and minerals, potentially posing the risk of nutritional imbalances and deficiencies such as skin and thyroid problems [1,2]. Furthermore, although the production of raw pet food is subjected to strict microbiological hygiene criteria (the EU animal by-products regulations 1069/2009 and 142/2011), pasteurization is by nature not undertaken, raising questions regarding bacterial contamination [3–5]. Concerns, raised by the American Veterinary Medical Association (AVMA) [1] and the Canadian Veterinary Medical Association (CVMA) [6], are based on evidence that raw food diets may cause disease in pets, as exemplified by the reported cases of salmonellosis in cats and dogs [7–9], yersiniosis in cats and dogs [10] and a case of brucellosis in a dog fed raw hare carcasses [11]. RMBDs have also been shown to be a significant source of *Salmonella* in healthy dogs and cats that consequently shed the organism at higher rates than animals fed conventional diets [10,12], with implications concerning public health. Case reports of human illness associated with pathogens in RMBDs are still scarce, and illnesses are probably under-reported [12]. However, four cases of an ongoing outbreak of *Salmonella* Reading in the USA were linked to raw pet food [13], and recently, an outbreak due to Shiga toxin producing *Escherichia coli* (STEC) O157:H7 in the UK was attributed to exposure to contaminated raw pet food [14]. Hence, there is growing evidence that pathogens occurring in RMBDs pose a risk of infectious disease to humans not only during handling of the feed and the feeding equipment, but also through the contamination of household surfaces and through close contact to the dogs and their faeces [3,4,15].

RMBDs have also been identified as a risk factor for the shedding of antimicrobial-resistant (AMR) bacteria in pets [16–21]. This is of particular concern, since antimicrobial resistance is currently one of the most pressing threats to human and animal health worldwide, affecting humans, animals and the environment [22]. Because of the use and overuse of antimicrobial agents in livestock production, food-producing animals have emerged as an important reservoir for antimicrobial resistance [23]. Accordingly, raw meat sold at retail level (beef, poultry and fish) has been identified as a major source of exposure of humans to AMR bacteria, including Enterobacteriaceae with resistance to drugs categorized by the World Health Organization (WHO) as critically important antimicrobial agents (CIAs) [24,25]. Correspondingly, RMBDs containing by-products from livestock slaughter represent a source of AMR and a potential health risk to animals and humans. Previous studies have identified the occurrence of *E. coli* isolates from RMBDs possessing the same resistance mechanisms as isolates from poultry, cattle and pigs, including one of the most important mechanisms of antimicrobial resistance in Enterobacteriaceae, the production of plasmid-mediated extended-spectrum β-lactamases (ESBLs) [17,26]. These findings are of great concern, since the occurrence of ESBL producers and other AMR bacteria may not only entail treatment failure in diseased companion animals, but also poses a potential health hazard to humans, either through direct transmission of AMR bacteria from animals to humans or indirectly through transmission of resistance genes [27]. Currently, the most prevalent ESBLs belong to the CTX-M type, with CTX-M-15 globally the most common among human clinical isolates [28].

The growing popularity of RMBDs and the concomitant potential risk to animal and human health prompted us to assess the microbiological quality of RMTBs with the focus on the molecular characterization of ESBL producers and other AMR Enterobacteriaceae isolated from commercially available raw pet food in Switzerland.

# 2. Material and methods

## 2.1. Sample collection

During September and October 2018, 47 RMBDs were commercially purchased by the investigators in pet shops in six cities within a radius of 300 km of the laboratory or via Internet shops. Four further samples were obtained from a small Swiss RMBD producing enterprise which was officially certified based on

hazard analysis and critical control points (HACCP) hygiene standards through the responsible county veterinary office.

Only samples that contained uncooked meat or organs that had not undergone any treatment, such as pasteurization or drying, were purchased. Only RMBDs intended for dogs were included.

Samples were transported in bags containing cold chain coolants and stored at −20°C. Before analysis, the samples were defrosted at 4°C. For all samples, analysis was performed before the provided 'use by' date.

Products were categorized into those originating from of beef cattle, poultry, horse, lamb, game, rabbit and fish. Types of meat within these categories included beef (including rumen) ($n = 15$), chicken ($n = 6$), horse ($n = 6$), lamb ($n = 6$), turkey ($n = 4$), rabbit ($n = 3$), salmon ($n = 3$), deer ($n = 2$), duck ($n = 1$), moose ($n = 1$), ostrich ($n = 1$), pangasius fish ($n = 1$), quail ($n = 1$) and reindeer ($n = 1$). Thirty-one samples contained meat originating from Switzerland, and 20 samples contained meat imported from Germany.

## 2.2. Quantitative bacteriology

For the examination for aerobic mesophilic bacteria (AMB) and Enterobacteriaceae, a subset of 10 g from each sample was placed in a sterile blender bag and homogenized for 60 s at a 1 : 10 ratio in buffered peptone water (BPW, Oxoid, Basingstoke, UK). The homogenates and decimal dilutions thereof in 0.9% NaCl were streaked onto plate count (PC) agar (Oxoid, Pratteln, Switzerland) for AMB and onto violet red bile glucose (VRBG) agar (Bio-Rad Laboratories AG, Reinach, Switzerland) for Enterobacteriaceae. PC plates were incubated for 72 h at 30°C under aerobic conditions and VRBG plates for 48 h at 37°C under anaerobic conditions. The number of colonies was counted to calculate the number of colony-forming units (cfu) per gram of raw pet food sample. Samples were rated based on the EU hygiene criteria that limit Enterobacteriaceae to $5 \times 10^3$ cfu g$^{-1}$ for raw meat intended for pet food production.

## 2.3. Detection of *Salmonella*

Examination for *Salmonella* spp. was done using a two-step enrichment procedure. Of each sample, 10 g was pre-enriched for 24 h at 37°C in 100 ml of BPW. From the first enrichment, 0.1 ml was incubated for 24 h at 42°C in 10 ml of Rappaport Vassiliadis (RV) broth (Oxoid, Pratteln, Switzerland). After plating 0.1 ml onto Rapid'*Salmonella* (RSal) medium agar (BioRad, Hercules, California, USA), plates were incubated for 24 h at 37°C. All typical colonies with violet morphologies were subjected to serological identification according to the Kauffmann–White–LeMinor scheme [29].

## 2.4. Selective isolation of antimicrobial-resistant Enterobacteriaceae

A subset of 10 g of sample was homogenized at a 1 : 10 ratio in Enterobacteriaceae enrichment (EE) broth (BD, Franklin Lakes, USA) for 24 h at 37°C. For the detection of ESBL-producing Enterobacteriaceae, one loopful of each of the enrichment cultures was streaked onto Brilliance ESBL agar plates (Oxoid, Hampshire, UK). Plates were incubated under aerobic conditions for 24 h at 37°C. All colonies with different chromaticity and morphology were subcultured on Brilliance ESBL agar plates for 24 h at 37°C to obtain pure cultures. Species were identified using matrix-assisted laser desorption ionization-–time-of-flight mass spectrometry (MALDI-TOF-MS, Bruker Daltronics, Bremen, Germany).

Colistin-resistant Gram-negative bacteria were selected by plating one loopful of the enriched cultures onto Luria-Bertani (LB, Difco Laboratories, Franklin Lakes, NJ, USA) agar plates containing 4 mg l$^{-1}$ colistin, 10 mg l$^{-1}$ vancomycin and 5 mg l$^{-1}$ amphotericin B. Plates were incubated under aerobic conditions for 24 h at 37°C. Colonies were subcultured on a selective medium containing 4 mg l$^{-1}$ colistin and each pure culture was identified using MALDI-TOF-MS. Species with intrinsic resistance to polymyxins (*Serratia marcescens*, *Proteus* spp., *Providencia* spp. and *Morganella* spp.) were discarded.

Screening for aminoglycoside-resistant Gram-negative bacteria was conducted using LB agar containing 200 mg l$^{-1}$ amikacin, 10 mg l$^{-1}$ vancomycin and 5 mg l$^{-1}$ amphotericin B, as described previously [30]. Colonies were subcultured on a selective medium containing 200 mg l$^{-1}$ amikacin and each pure culture was identified using MALDI-TOF-MS.

## 2.5. Antimicrobial susceptibility testing

Antimicrobial susceptibility testing (AST) was performed using the disc-diffusion method according to the guidelines of the Clinical and Laboratory Standards Institute (CLSI) [31], and the antibiotics ampicillin (AM), amoxicillin with clavulanic acid (AMC), azithromycin (AZM), cefazolin (CZ), cefepime (FEP), cefotaxime

(CTX), chloramphenicol (C), ciprofloxacin (CIP), fosfomycin (FOS), gentamicin (G), kanamycin (K), nalidixic acid (NA), nitrofurantoin (F/M), streptomycin (S), sulfamethoxazole/trimethoprim (SXT) and tetracycline (TE) (Becton Dickinson, Allschwil, Switzerland). Results were interpreted according to CLSI performance standards [31]. For azithromycin, an inhibition zone of less than or equal to 12 mm was interpreted as resistant. For isolates harbouring *mcr-1*, the minimum inhibitory concentration (MIC) of colistin was determined by broth microdilution according to the European Committee on Antimicrobial Susceptibility Testing (EUCAST), 2019 (eucast.org). The results were interpreted according to the breakpoints suggested by EUCAST for Enterobacterales (susceptible, MIC $\leq 2$ mg $l^{-1}$; resistant, MIC $> 2$ mg $l^{-1}$).

Isolates displaying resistance to three or more classes of antimicrobials were defined as multidrug-resistant (MDR), as proposed by Magiorakos *et al.* [32], counting β-lactams as one class.

## 2.6. Detection of antimicrobial-resistant genes

DNA was extracted by a standard heat lysis protocol and analysed by PCR for the presence of AMR genes. Synthesis of primers and direct DNA sequencing were carried out by Microsynth (Balgach, Switzerland). Purification of amplification products was performed using a PCR purification kit (Qiagen, Courtaboeuf, France). Nucleotide sequences were analysed with CLC Main Workbench 6.6.1. Database searches were performed using the BLASTN program of NCBI (http://www.ncbi.nlm.nih.gov/blast/).

The identification of *bla*$_{ESBL}$ genes was established, as described previously [33–35].

Screening by PCR for *mcr-1* through *mcr-5* was undertaken, as described by Rebelo *et al.* [36].

Presumptive 16S rRNA methylase producers were analysed for the presence of *armA*, *rmtA*, *rmtB*, *rmtC* and *rmtD*, as described previously [37].

## 2.7. Phylogenetic characterization of *E. coli* isolates

Phylogenetic classification of the *E. coli* isolates into one of the eight groups A, B1, B2, C, D, E, F (*E. coli sensu stricto*) or *Escherichia* clade I, was performed, as described by Clermont *et al.* [38].

## 2.8. Multilocus sequence typing of *E. coli* and *Klebsiella pneumoniae*

For multilocus sequence typing (MLST) of *E. coli* isolates, internal fragments of the seven housekeeping genes (*adk*, *fumC*, *gyrB*, *icd*, *mdh*, *purA* and *recA*) were amplified by PCR, as described by Wirth *et al.* [39]. Sequences were imported into the *E. coli* MLST database website (https://enterobase.warwick.ac.uk) to determine MLST types and clonal complexes (CCs).

MLST of the *K. pneumoniae* isolates was performed by the amplification and sequencing of the seven housekeeping genes *gapA*, *infB*, *mdh*, *pgi*, *phoE*, *rpoB* and *tonB*, according to the previously described methods [40]. Sequence types were determined, according to the MLST database (https://bigsdb.pasteur.fr).

# 3. Results

## 3.1. Bacterial analysis

An overview of the distribution of the AMB counts and Enterobacteriaceae counts for the different types of meat is given in table 1.

Among the 51 raw pet food samples analysed in this study, the AMB count ranged from $8.2 \times 10^4$ to $7.4 \times 10^8$ cfu $g^{-1}$ (median value $8.8 \times 10^6$ cfu $g^{-1}$). Overall, 28 (55%) of the products scored higher than $5 \times 10^6$ cfu AMB $g^{-1}$ meat.

Enterobacteriaceae PCs ranged from $6 \times 10^2$ to $2.2 \times 10^7$ cfu $g^{-1}$ (median value $4.1 \times 10^4$ cfu $g^{-1}$). For 37 (72.5%) of the diets, the score exceeded $5 \times 10^3$ cfu $g^{-1}$. Based on the EU regulations, these 37 products did not meet the hygiene criteria of less than $5 \times 10^3$ cfu $g^{-1}$ for raw meat intended for pet food production (figure 1*a*). Diets exceeding this limit varied between the suppliers, from 50% of the diets from supplier H to 100% of the diets from suppliers C and D (figure 1*b*).

*Salmonella* species were isolated from two (3.9%) RMBDs. The serotypes were identified as monophasic *Salmonella* Typhimurium 4,12:i:- isolated from RMBDs containing lamb (16.7% of the lamb meat samples), and *Salmonella* London isolated from turkey (25% of the turkey meat samples). Products contaminated with *Salmonella* originated from suppliers G and F.

**Table 1.** Distribution of aerobic mesophilic count and Enterobacteriaceae count for RMBDs for pets containing different types of meat. AMBC, aerobic mesophilic count; cfu, colony-forming unit; EC, Enterobacteriaceae; count; —, not applicable.

| type of meat | no. of samples | bacteria | no. of samples with cfu of | | | | | | median |
|---|---|---|---|---|---|---|---|---|---|
| | | | $10^2$–$<10^3$ | $10^3$–$<10^4$ | $10^4$–$<10^5$ | $10^5$–$<10^6$ | $10^6$–$<10^7$ | $>10^7$ | |
| beef | 15 | AMBC | 0 | 0 | 0 | 1 | 2 | 12 | $4.1 \times 10^7$ |
| | | EC | 0 | 1 | 4 | 7 | 2 | 1 | $2.3 \times 10^5$ |
| chicken | 6 | AMBC | 0 | 0 | 0 | 4 | 1 | 1 | $7.85 \times 10^5$ |
| | | EC | 1 | 1 | 2 | 1 | 1 | 0 | $2.55 \times 10^4$ |
| deer | 2 | AMBC | 0 | 0 | 0 | 0 | 0 | 2 | $1.4 \times 10^8$ |
| | | EC | 0 | 0 | 0 | 1 | 0 | 1 | $5.91 \times 10^6$ |
| duck | 1 | AMBC | 0 | 0 | 0 | 0 | 1 | 0 | — |
| | | EC | 0 | 0 | 1 | 0 | 0 | 0 | — |
| horse | 6 | AMBC | 0 | 0 | 0 | 1 | 2 | 3 | $1.45 \times 10^7$ |
| | | EC | 1 | 3 | 1 | 0 | 1 | 0 | $1.5 \times 10^3$ |
| lamb | 6 | AMBC | 0 | 0 | 0 | 1 | 3 | 2 | $2.3 \times 10^6$ |
| | | EC | 0 | 3 | 2 | 1 | 0 | 0 | $2.12 \times 10^4$ |
| moose | 1 | AMBC | 0 | 0 | 0 | 0 | 0 | 1 | — |
| | | EC | 0 | 0 | 1 | 0 | 0 | 0 | — |
| ostrich | 1 | AMBC | 0 | 0 | 0 | 0 | 0 | 1 | — |
| | | EC | 0 | 0 | 0 | 0 | 1 | 0 | — |
| pangasius | 1 | AMBC | 0 | 0 | 0 | 0 | 1 | 0 | — |
| | | EC | 0 | 1 | 0 | 0 | 0 | 0 | — |
| quail | 1 | AMBC | 0 | 0 | 0 | 1 | 0 | 0 | — |
| | | EC | 0 | 1 | 0 | 0 | 0 | 0 | — |

(Continued.)

**Table 1.** (*Continued.*)

| type of meat | no. of samples | bacteria | no. of samples with cfu of | | | | | | median |
|---|---|---|---|---|---|---|---|---|---|
| | | | $10^2$–$<10^3$ | $10^3$–$<10^4$ | $10^4$–$<10^5$ | $10^5$–$<10^6$ | $10^6$–$<10^7$ | $>10^7$ | |
| rabbit | 3 | AMBC | 0 | 0 | 0 | 2 | 1 | 0 | $8.3 \times 10^5$ |
| | | EC | 1 | 2 | 0 | 0 | 0 | 0 | $10^3$ |
| reindeer | 1 | AMBC | 0 | 0 | 0 | 0 | 0 | 1 | — |
| | | EC | 0 | 0 | 1 | 0 | 0 | 0 | — |
| salmon | 3 | AMBC | 0 | 0 | 1 | 0 | 1 | 1 | $8.8 \times 10^6$ |
| | | EC | 2 | 0 | 1 | 0 | 0 | 0 | $8.0 \times 10^2$ |
| turkey | 4 | AMBC | 0 | 0 | 0 | 1 | 2 | 1 | $1.65 \times 10^6$ |
| | | EC | 1 | 1 | 2 | 0 | 0 | 0 | $1.74 \times 10^4$ |

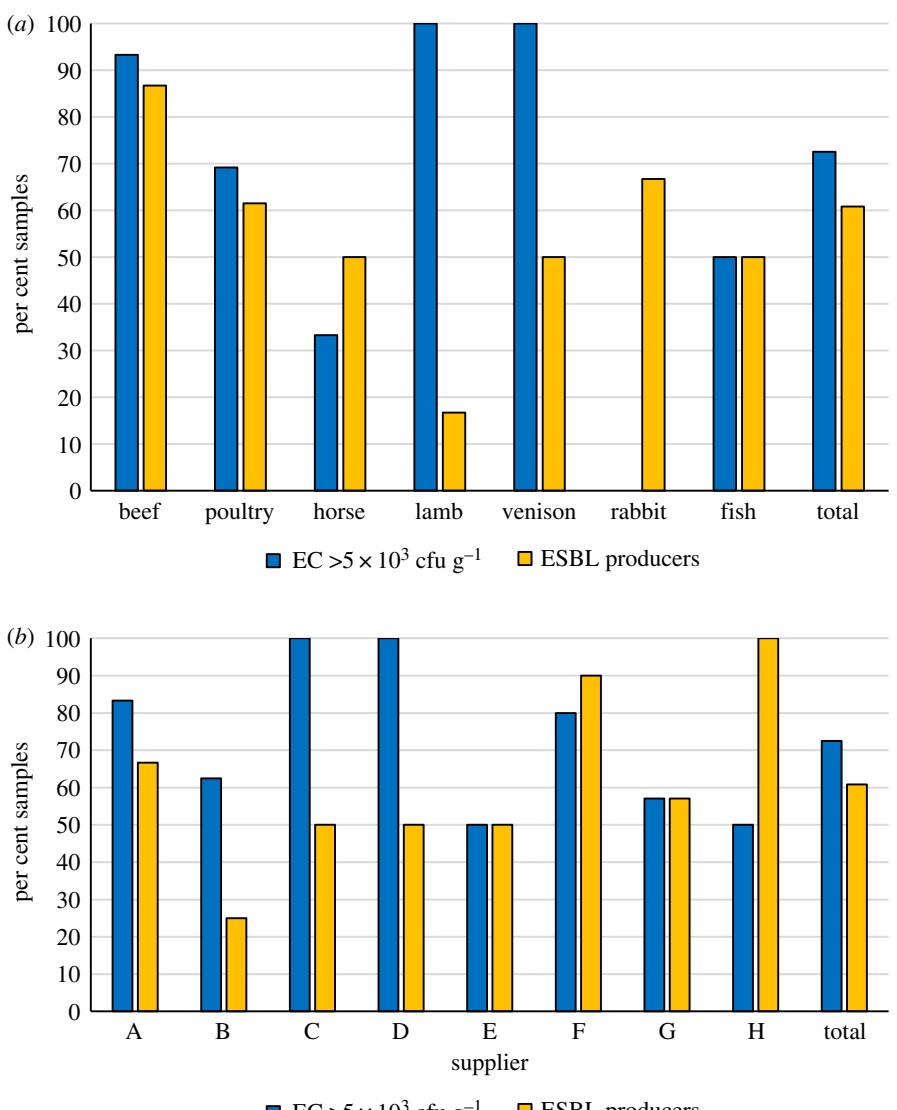

**Figure 1.** Microbiological quality of different categories of 51 samples of RMBDs for pets. (*a*) Per cent of RMBDs exceeding the EU hygiene criterion of $5 \times 10^3$ cfu g$^{-1}$ for raw meat intended for pet food production, and per cent of RMBDs contaminated with ESBL-producing Enterobacteriaceae. (*b*) Per cent of 51 RMBDs of eight different suppliers A–H containing Enterobacteriaceae counts that exceed $5 \times 10^3$ cfu g$^{-1}$ and per cent of RMBDs of eight suppliers contaminated with ESBL-producing Enterobacteriaceae.

## 3.2. Prevalence of antimicrobial-resistant Enterobacteriaceae in RMBDs

The types of meat containing Enterobacteriaceae-harbouring plasmid-mediated antimicrobial resistance, their origins and the number of isolates per sample are shown in figure 2. In total, AMR bacteria were found in 32 (62.7%) of the samples, and 14 (27.5%) contained more than one distinct AMR isolate.

ESBL-producing Enterobacteriaceae were isolated from 31 (60.8%) RMBDs. They included 13 (86.7%) of the 15 samples containing offal from cattle such as muscle meat, blood, fat and rumen, eight (61.5%) of the 13 poultry-based diets, three (50%) of the six horse meat samples, one (16.7%) of the lamb meat diets, two (50%) samples of venison, two of three (66.7%) of the rabbit meat samples and two (50%) RMBDs containing fish (figure 1*a*). Fourteen samples yielded two or more distinct isolates (figure 2). RMBDs contaminated with ESBL producers were detected in products from all eight suppliers, with variations between 25% of all products from supplier B and 100% of the products from supplier H (figure 1*b*).

*Escherichia coli* harbouring the colistin-resistance gene *mcr-1* were identified in two (3.9%) samples. One sample contained offal of horse, and one minced quail meat (figure 2).

*Citrobacter freundii* harbouring the plasmid-mediated aminoglycoside-resistant gene *rmtB* was detected in one (2%) sample which consisted of rabbit muscle meat (figure 2).

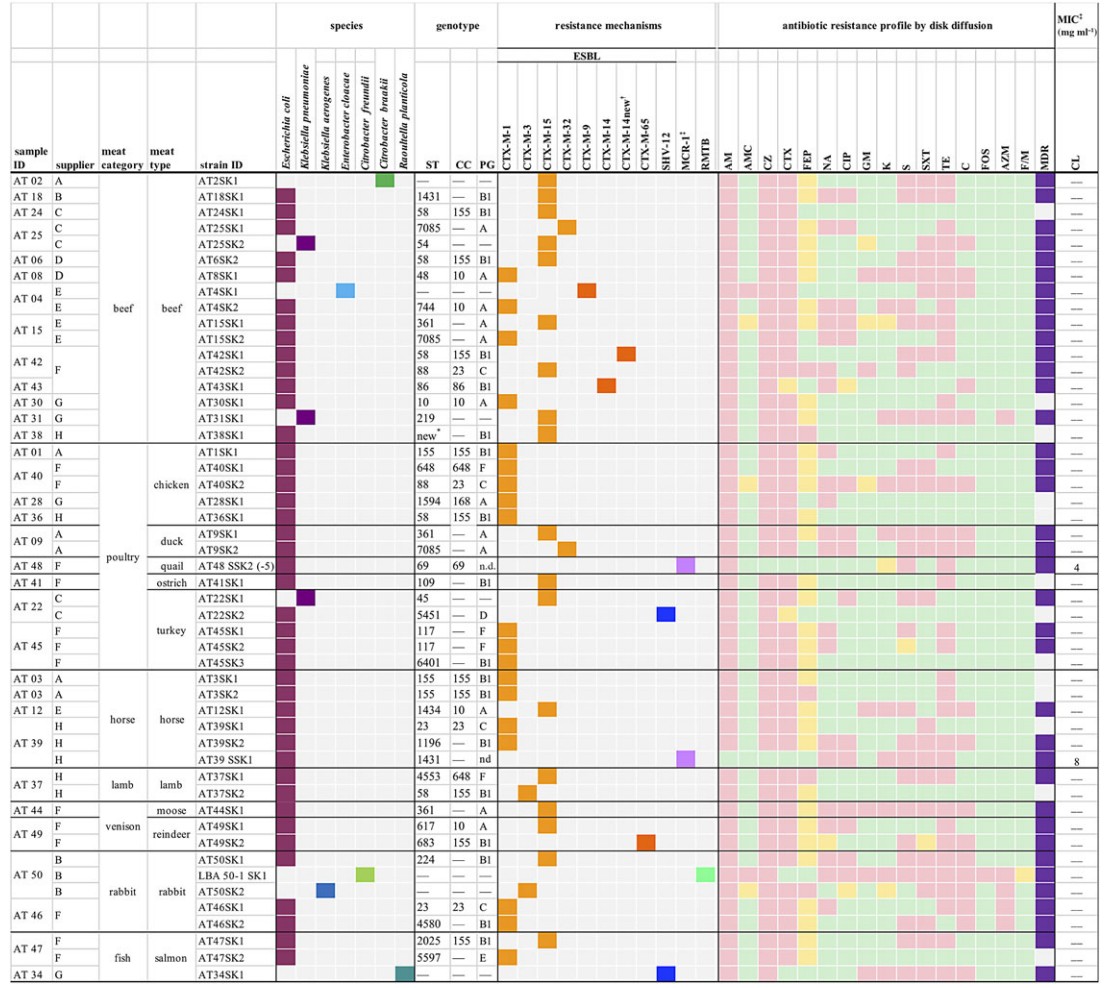

**Figure 2.** Source data and characteristics of ESBL-, MCR-1- and RMTB-producing Enterobacteriaceae isolated from RMBDs for pets. Colours of squares categorizing ESBLs: light orange, CTX-M-group 1; dark orange, CTX-M-group 9; blue, SHV enzymes. Colours categorizing antibiotic resistance profiles: pink, resistant; yellow, intermediate; green, susceptible. AM, ampicillin; AMC, amoxicillin-clavulanic acid; AZM, aztreonam; C, chloramphenicol; CC, clonal complex; CL, colistin; CZ, cefazolin; CTX, cefotaxime; CIP, ciprofloxacin; FEP, cefepime; F/M, nitrofurantoin; FOS, fosfomycin; GM, gentamicin; K, kanamycin; MDR, multidrug resistance; NA, nalidixic acid; PG, phylogenetic group; S, streptomycin; SXT, sulfamethoxazole/trimethoprim; ST, sequence type; TE, tetracycline; n.d., not determined; —, not applicable. *New sequence type with the allelic profile *adk* (1), *fumC* (1120), *gyrB* (44), *icd* (9), *mdh* (11), *purA* (9) and *recA* (7). †New CTX-M-14 variant with aminoacid substitution P180 → L. ‡The susceptibility test results for colistin were interpreted according to the susceptibility and resistance clinical breakpoints suggested by the EUCAST (European Committee on Antimicrobial Susceptibility Testing, 2019) for Enterobacterales (susceptible, MIC $\leq$ 2 mg l$^{-1}$; resistant, MIC > 2 mg l$^{-1}$).

## 3.3. Identification of ESBL-producing Enterobacteriaceae and of $bla_{ESBL}$ genes

In total, 47 ESBL producers were retrieved. Of these, 40 (85.1%) were identified as *E. coli*, three (6.4%) were classified as *Klebsiella pneumoniae* and one (2.1%) each were *Klebsiella aerogenes* (formerly *Enterobacter aerogenes*) [41], *Enterobacter cloacae*, *Citrobacter braakii* and *Raoultella planticola* (figure 2).

All 47 isolates were characterized with regard to their ESBL genotypes. In total, $bla_{CTX-M}$ genes were detected in 45 (95.7%) of the strains. The vast majority of 41 (91.1%) of the $bla_{CTX-M}$ genes belonged to CTX-M-group 1, and four (8.9%) to CTX-M-group 9. Two strains harboured $bla_{SHV-12}$ (figure 2).

Of the 40 ESBL-producing *E. coli* isolates, 19 (47.5%) harboured $bla_{CTX-M-1}$, 14 (35%) carried $bla_{CTX-M-15}$ and two (5%) tested positive for $bla_{CTX-M-32}$. The remaining strains occurred as single isolates (2.5% each), harbouring $bla_{CTX-M-3}$, $bla_{CTX-M-14}$, a new variant of $bla_{CTX-M-14}$, $bla_{CTX-M-65}$ and $bla_{SHV-12}$, respectively (figure 2). Of the three *K. pneumoniae* isolates, all (100%) contained $bla_{CTX-M-15}$ (figure 2). The *E. cloacae* strain harboured $bla_{CTX-M-9}$, the *K. aerogenes* carried $bla_{CTX-M-3}$, *C. braakii* tested positive for $bla_{CTX-M-15}$ and *R. planticola* harboured $bla_{SHV-12}$ (figure 2).

Regarding the distribution of the most frequently identified $bla_{ESBL}$ genes, $bla_{CTX-M-1}$ was detected in eight (61.5%) of the 13 strains isolated from RMBDs containing poultry, in four (23.5%) of the 17 isolates from beef and in four (80%) of the ESBL producers isolated from horse meat. By contrast, $bla_{CTX-M-15}$ was identified in nine (52.9%) of the 17 strains originating from diets containing beef, in three (23.1%) strains isolated from poultry meat-based diets and in one (20%) ESBL producer isolated from horse meat.

## 3.4. Antimicrobial susceptibility patterns

Resistance profiles were determined for a total of 50 isolates, including 47 ESBL producers, two MCR-1 producing *E. coli* and one RMTB-producing *C. braakii* (figure 2). Overall, the resistance to cefotaxime and cefepime was observed for 44 (88%) and five (10%) of the isolates. AST for other classes of antimicrobials revealed that 21 (42%) isolates were resistant to the quinolone antibiotic nalidixic acid and 15 (30%) were resistant to the fluoroquinolone ciprofloxacin. Resistance to aminoglycosides was detected in six (12%) isolates resistant to gentamicin, 10 (20%) resistant to kanamycin and 27 (54%) resistant to streptomycin. Resistance to the folate pathway inhibitor sulfamethoxazole/trimethoprim was found in 25 (50%) isolates. Tetracycline resistance was noted in 36 (72%) and chloramphenicol resistance in 18 (36%) isolates. Furthermore, resistance to fosfomycin and azithromycin was observed in one (2%) and five (10%) of the isolates. None of the isolates were resistant to nitrofurantoin.

MDR was detected in 37 (74%) of the isolates (figure 2). Colistin MIC values were 8 mg l$^{-1}$ for strain AT39 SSK1 and 4 mg l$^{-1}$ for strain AT48 SSK2 (figure 2).

## 3.5. Genotypic characteristics of *E. coli* and *K. pneumoniae* isolates

MLST of all 42 *E. coli* isolates identified 28 different sequence types, including one new sequence type with the allelic profile *adk* (1), *fumC* (1120), *gyrB* (44), *icd* (9), *mdh* (11), *purA* (9) and *recA* (7).

Twenty-one (50%) strains belonged to a collective of STs that occurred only once or twice, followed by 10 (23.8%) belonging to CC155 (ST155, ST58 and closely related STs), five (11.9%) CC10 (ST10 and related STs) and CC23 (four strains). Three (7.1%) strains typed ST361 (figure 2).

Phylogenetic typing was performed for the 40 ESBL-producing *E. coli* and allocated 30 (75%) of the isolates to group A or B1, which typically contain commensal *E. coli* strains. Four (10%) belonged to phylogroup C. Five (12.5%) belonged to extraintestinal pathogenic phylogroups D and F, and one belonged to phylogroup E. None of the isolates belonged to extraintestinal pathogenic phylogroup B2.

MLST of the *K. pneumoniae* strains detected ST45, ST54 and ST219 (figure 2).

# 4. Discussion

Overall, the microbiological quality of the RMBDs analysed in this study was unsatisfactory for 72.5% of the products with regard to the EU hygiene criteria for the raw meat intended for pet food production, irrespective of the supplier. The microbiological quality was also lower than recently reported for comparable products in The Netherlands [4]. The current lack of comparative data from other countries and the low sample size in our study prevent a conclusive evaluation of the microbiological quality of RMBDs. In spite of these limitations, our data contribute to the growing evidence that RMBDs constitute a hygiene hazard.

While raw food diets are produced with minimal guidance in the USA and Canada [12], the EU regulations 1069/2009 and 142/2011 lay down an Enterobacteriaceae limit of $5 \times 10^3$ cfu g$^{-1}$ for by-products of slaughtered animals intended for pet food. The majority (72.5%) of the RMBDs in this study exceeded this threshold, and the prevalence of unacceptable products was thereby higher than the 52% recently reported in a Swedish study [3,42]. Notably, in our study, only one sample per batch was tested, precluding a definitive hygiene classification of any batch of RMBD. Nevertheless, high levels of Enterobacteriaceae in RMBDs potentially pose a health risk to animals and humans.

Pathogens, such as *Salmonella*, may cause disease in pets and contaminate the environment and humans with which the pets come into contact. In our study, *Salmonella* was isolated from 3.9% of the samples. This finding raises concern with regard to the zero tolerance policy for *Salmonella* laid down by the EU regulations 1069/2009 and 142/2011, as mentioned above. Previous investigators have reported the detection of *Salmonella* in 7% of RMBDs in Sweden and the USA [3] and 20% in The Netherlands and Canada [4]. The significance of these findings should not be underestimated, since *Salmonella* spp. pose a serious health risk to vulnerable individuals including small children, pregnant

women, immunocompromised persons and the elderly. Of the *Salmonella* serotypes identified in this study, monophasic *Salmonella* Typhimurium 4,12:i:- detected in lamb meat ranked among the three most commonly reported *Salmonella* serotypes associated with laboratory-confirmed cases of human salmonellosis in the EU and Switzerland in 2017 [43]. Monophasic *S.* 1,4,[5],12:i:- has also been responsible for foodborne outbreaks in Europe and worldwide [44]. By contrast, *Salmonella* London isolated from an RMBD containing turkey meat is not a common serotype, constituting only six (0.4%) of 1448 cases of human non-typhoidal *Salmonella* infection reported by the Swiss Federal Office of Public Health (SFOPH) in 2018 [45]. RMBDs may represent an important source of rare *Salmonella* serotypes of currently unclear pathogenicity, with implications for public health [46].

A further threat to public health is the global dissemination of AMR bacteria. Resistance to third-generation cephalosporins, e.g. cefotaxime, presents a massive limitation of options to treat infections caused by MDR Enterobacteriaceae [28]. Previously, two Dutch longitudinal studies reported associations between RMBDs and faecal carriage of ESBL-producing *E. coli* in cats and dogs [16,17]. In the studies of dogs in the UK, feeding dogs RMBDs, especially raw poultry, was identified as a risk factor for faecal ESBL-producing *E. coli* [20,21]. Accordingly, the high rate of contamination (60.8%) of RMBDs with ESBL producers, as well as the very high rate (74%) of MDR among the Enterobacteriaceae detected in this study is of great concern, although this rate is slightly lower than results from similar studies from The Netherlands that reported higher prevalences (77.8 and 80%) of ESBL producers in RMBDs [4,17].

The most frequently detected ESBLs in this study were CTX-M-1 (40.2%) and CTX-M-15 (38.3%). CTX-M-1 is widespread in livestock and the food chain in Europe [28]. By contrast, CTX-M-15 is globally the most prevalent variant among human clinical ESBL-producing isolates worldwide, but infrequent in samples from livestock and food in European countries [47,48]. However, there is evidence that CTX-M-15 producers are more prevalent among faecal samples of cattle than those of other livestock, at least in some countries, including the UK and Switzerland [49,50], which may account for the fact that we detected the majority (52.9%) of the CTX-M-15 producers in RMBDs containing beef. The possibility of transmission of CTX-M-15-producing isolates from RMBDs to humans deserves further attention. Notably, plasmid analysis, including transfer assays and molecular plasmid typing, was not performed in this study, which limits our ability to fully delineate the epidemiologic features of the ESBL-producing Enterobacteriaceae identified in RMBDs.

Less frequent ESBL types found in this study include CTX-M-32, a structural variant of CTX-M-1 which has also been found in *E. coli* isolated from raw cat pet food and from faeces of cats fed RMBDs [17] and CTX-M-14, which, while found increasingly in human clinical isolates in Europe, is predominant in Asia [28]. In addition, we identified a novel CTX-M-14 variant carrying a Prolin180 → Lysin substitution.

The predominance of commensal *E. coli* and the diversity of sequence types among the ESBL-producing *E. coli* and *K. pneumoniae* indicate that the majority of the RMBD-associated ESBL producers may represent an indirect threat to public health predominantly through colonization of the human and animal gut [51]. While harmless for healthy individuals, faecal colonization is a risk factor for infections with ESBL producers [51]. The most prominent CCs identified in the current study have been described internationally among ESBL producers from various sources. For example, ESBL-producing *E. coli* ST58 and ST155 (CC155) are described globally from a wide range of sources, including healthy humans, livestock and wildlife [52–55].

Furthermore, *E. coli* belonging to international clones CC10 and CC23 associated with CTX-M-1 and CTX-M-15, have been isolated from healthy humans, livestock and meat, as well as from the environment [53]. In particular, ST10, ST23 and ST155 are frequently observed among avian pathogenic *E. coli* (APEC) [56].

Several STs less frequently identified in this study have also been described among ESBL-producing *E. coli* isolated from diseased companion animals and livestock. For instance, among CTX-M-1 producing *E. coli* isolated from the urine of diseased dogs [57] and among CTX-M-2-producing bovine mastitis isolates [58]. Furthermore, *E. coli* ST117, isolated from turkey meat in this study, has been identified among APEC strains associated with increased mortality and colibacillosis in broilers [59,60].

Notably, we detected two *E. coli* isolates belonging to CC648 which is an international high-risk multidrug-resistant clone that has emerged among companion animals in Europe [61]. Furthermore, *K. pneumoniae*, detected in three RMBDs in this study, may cause a wide range of nosocomial and community-acquired infections in humans and in companion animals, including pneumonia, urinary tract infection (UTI) and bacteraemia [62].

Our results suggest that RMBDs of the types analysed in this study represent a hitherto underappreciated source of ESBL-producing Enterobacteriaceae.

Moreover, two RMBD samples were contaminated with *E. coli* harbouring the plasmid-mediated colistin resistance gene *mcr-1*. Colistin has become a crucial last resort antimicrobial to treat

infections caused by MDR Gram-negative bacteria [24]. MCR-1 producers have been isolated from food animals, raw meat and human samples [63], but to our knowledge, their occurrence in commercially available RMBDs has not been documented before. It is also particularly alarming that one of the *mcr-1* harbouring *E. coli* isolates belonged to the pandemic clonal lineage ST69 which is associated with community-acquired and healthcare-associated urinary tract infections (UTIs) worldwide [64].

## 5. Conclusion

RMBDs represent an emerging route of exposure of pets and their owners to bacterial pathogens such as *Salmonella* and to MDR Enterobacteriaceae. Resistant bacteria occurring in RMBDs include those harbouring $bla_{ESBL}$ genes that are identical to those in ESBL producers causing disease in animals and humans worldwide, and those resistant to crucially important antimicrobials such as aminoglycosides and colistin.

The possibility of transmission of *Salmonella* and of MDR Enterobacteriaceae from RMBDs to companion animals and their owners poses a severe health risk, particularly to vulnerable persons such as infants, the elderly, pregnant or immunocompromised individuals. Appropriate measures, such as activities that raise the awareness of antimicrobial resistance from the pet food safety perspective and providing information to pet owners on the correct handling of RMBDs, should be established in order to reduce the risk and ensure animal and public health.

Data accessibility. The datasets supporting this article have been uploaded as part of the electronic supplementary material.
Authors' contributions. M.N.-I. conducted data analysis and interpretation and drafted the manuscript; A.T. collected the samples, performed laboratory tests and contributed to data analysis; K.Z. helped design the study and performed laboratory tests; R.S. designed and coordinated the study, participated in data analysis and critically revised the manuscript; all authors gave final approval for publication.
Competing interests. The authors declare no competing interests.
Funding. This work was partly supported by the Swiss Federal Office of Public Health, Division Communicable Diseases.
Acknowledgements. The authors thank Andrea Müller for serotyping the *Salmonella* isolates and Sabrina Püntener-Simmen and Kira Schmitt for their technical assistance.

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
