## [Reviewer comments · Royal Society Open Science]

Review History

RSOS-191170.R0 (Original submission)

Review form: Reviewer 1

Is the manuscript scientifically sound in its present form?

No

Are the interpretations and conclusions justified by the results?

Yes

Is the language acceptable?

Yes

Do you have any ethical concerns with this paper?

No

Have you any concerns about statistical analyses in this paper?

No

Recommendation?

Accept with minor revision (please list in comments)

Comments to the Author(s)

This is an interesting and relevant study of bacterial contamination of raw meat diets. There have been a few studies reporting an association between raw meat diets and cephalosporin resistant Enterobacteriaceae in dogs and cats. These should be mentioned/discussed as they directly pertain to the relevance of these data.

Any prevalence study needs detailed description of the sample population and sampling methodology. Very little is provided here. How were samples collected? Were all samples available in the store purchased? If not, how were they selected? Was sampling done in one city or more (and where)? Were they all different products? Did they all contain meat of Swiss origin? If not, what were the sources? Were any high pressure pasteurized or otherwise treated to reduce pathogens? Were all tested before any provided 'best before' or 'use by' dates?

How many colonies were tested per plate? How were they selected?

Line 235: Standard deviation is not the appropriate measure to report with median.

Line 236/239: The EU regulation limits are probably best first reported in the methods.

Was consideration given to any basic statistical comparisons, such as the prevalence between animal sources?

Review form: Reviewer 2

Is the manuscript scientifically sound in its present form?

Yes

Are the interpretations and conclusions justified by the results?

Yes

Is the language acceptable?

Yes

Do you have any ethical concerns with this paper?

No

Have you any concerns about statistical analyses in this paper?

No

Recommendation?

Reject

Comments to the Author(s)

This new version has very little modification compared to the first one. I still think that this is a solid and nice descriptive work that would be more suitable as a short communication in a journal that is more veterinary oriented (Vet mic for example).

Minor comment:
Line 272: formerly instead of formally

Review form: Reviewer 3

Is the manuscript scientifically sound in its present form?

Yes

Are the interpretations and conclusions justified by the results?

Yes

Is the language acceptable?

Yes

Do you have any ethical concerns with this paper?

No

Have you any concerns about statistical analyses in this paper?

No

Recommendation?

Accept as is

Comments to the Author(s)

The manuscript was improved and corrections/suggestions were performed as requested. My final comment is that limitations related to ESBL plasmid characterization should be included, since this is also an important public health issue.

Decision letter (RSOS-191170.R0)

28-Aug-2019

Dear Dr Nüesch-Inderbinen

On behalf of the Editors, I am pleased to inform you that your Manuscript RSOS-191170 entitled "Raw meat-based diets for companion animals: a potential source of transmission of pathogenic and antimicrobial resistant Enterobacteriaceae" has been accepted for publication in Royal Society Open Science subject to minor revision in accordance with the referee suggestions. Please find the referees' comments at the end of this email.

The reviewers and handling editors have recommended publication, but also suggest some minor revisions to your manuscript. Therefore, I invite you to respond to the comments and revise your manuscript.

• Ethics statement

If your study uses humans or animals please include details of the ethical approval received, including the name of the committee that granted approval. For human studies please also detail

whether informed consent was obtained. For field studies on animals please include details of all permissions, licences and/or approvals granted to carry out the fieldwork.

- Data accessibility

If you wish to submit your supporting data or code to Dryad (<http://datadryad.org/>), or modify your current submission to dryad, please use the following link:
<http://datadryad.org/submit?journalID=RSOS&manu=RSOS-191170>

- Competing interests

- Authors' contributions

- Acknowledgements

- Funding statement

Because the schedule for publication is very tight, it is a condition of publication that you submit

the revised version of your manuscript before 06-Sep-2019. Please note that the revision deadline will expire at 00.00am on this date. If you do not think you will be able to meet this date please let me know immediately.

If your manuscript is newly submitted and subsequently accepted for publication, you will be

asked to pay the article processing charge, unless you request a waiver and this is approved by Royal Society Publishing. You can find out more about the charges at <http://rsos.royalsocietypublishing.org/page/charges>. Should you have any queries, please contact openscience@royalsociety.org.

on behalf of Prof Kevin Padian (Subject Editor)
openscience@royalsociety.org

Editor comments:

Thanks for your revisions. Based on the reviewers' comments we will accept the manuscript with the provision that you attend to the several remaining comments that they have. Best wishes for your final version.

Reviewer comments to Author:

Reviewer: 1

Comments to the Author(s)

This is an interesting and relevant study of bacterial contamination of raw meat diets. There have been a few studies reporting an association between raw meat diets and cephalosporin resistant Enterobacteriaceae in dogs and cats. These should be mentioned/discussed as they directly pertain to the relevance of these data.

Any prevalence study needs detailed description of the sample population and sampling methodology. Very little is provided here. How were samples collected? Were all samples available in the store purchased? If not, how were they selected? Was sampling done in one city or more (and where)? Were they all different products? Did they all contain meat of Swiss origin? If not, what were the sources? Were any high pressure pasteurized or otherwise treated to reduce pathogens? Were all tested before any provided 'best before' or 'use by' dates?

How many colonies were tested per plate? How were they selected?

Line 235: Standard deviation is not the appropriate measure to report with median.

Line 236/239: The EU regulation limits are probably best first reported in the methods.

Was consideration given to any basic statistical comparisons, such as the prevalence between animal sources?

Reviewer: 2

Comments to the Author(s)

This new version has very little modification compared to the first one. I still think that this is a solid and nice descriptive work that would be more suitable as a short communication in a journal that is more veterinary oriented (Vet mic for example).

Minor comment:

Line 272: formerly instead of formally

Reviewer: 3

Comments to the Author(s)

The manuscript was improved and corrections/suggestions were performed as requested. My final comment is that limitations related to ESBL plasmid characterization should be included, since this is also an important public health issue.

Author's Response to Decision Letter for (RSOS-191170.R0)

See Appendix A.

Decision letter (RSOS-191170.R1)

18-Sep-2019

Dear Dr Nüesch-Inderbinen,

I am pleased to inform you that your manuscript entitled "Raw meat-based diets for companion animals: a potential source of transmission of pathogenic and antimicrobial resistant Enterobacteriaceae" is now accepted for publication in Royal Society Open Science.

Kind regards,

Lianne Parkhouse
Royal Society Open Science
openscience@royalsociety.org

on behalf of the Associate Editor and Professor Kevin Padian (Subject Editor)
openscience@royalsociety.org

Appendix A

Response to Referees of Manuscript RSOS-191170, Nüesch-Inderbinnen et al., "Raw meat-based diets for companion animals: a potential source of transmission of pathogenic and antimicrobial resistant Enterobacteriaceae"

Reviewer: 1

Comments to the Author(s)

Comment #1

This is an interesting and relevant study of bacterial contamination of raw meat diets. There have been a few studies reporting an association between raw meat diets and cephalosporin resistant Enterobacteriaceae in dogs and cats. These should be mentioned/discussed as they directly pertain to the relevance of these data.

REPLY:

Thank you for your comment and the opportunity to update our manuscript. We have amended the introduction by mentioning two further studies (Baede et al., 2015; Wedley et al, 2017), as well as a recently published review (Davies, 2019), to those already listed:

Line 99: RMBDs have also been identified as a risk factor for the shedding of antimicrobial resistant bacteria in pets (16- 21).

In addition, the discussion has been extended by specifically mentioning the studies that identified an association of RMBDs and the presence of ESBL producers in cats and dogs (Baede et al., 2015; Wedley et al, 2017):

Line 370: Previously, two Dutch longitudinal studies reported associations between RMBDs and faecal carriage of ESBL producing *E. coli* in cats and dogs (16, 17). In studies of dogs in the UK, feeding dogs RMBDs, especially raw poultry, was identified as a risk factor for faecal ESBL producing *E. coli* (20, 21).

new References:

Line 519: 16. Baede V O, Wagenaar J A, Broens E M et al. Longitudinal study of extended-spectrum- β -lactamase- and AmpC-producing Enterobacteriaceae in household dogs. *Antimicrob Agents Chemother.* 2015;59:3117-3124.

Line 526: 18. Davies R H, Lawes J R Wales A D. Raw diets for dogs and cats: a review, with particular reference to microbiological hazards. *J Small Anim Pract.* 2019;60:329-339.

Line 534: 21. Wedley A L, Dawson S, Maddox T W et al. Carriage of antimicrobial resistant *Escherichia coli* in dogs: Prevalence, associated risk factors and molecular characteristics. *Vet Microbiol.* 2017;199:23-30.

Comment #2

Any prevalence study needs detailed description of the sample population and sampling methodology. Very little is provided here.

REPLY:

We agree that there is a lack of information on the sampling methodology. To provide a more detailed description, we have addressed your queries and made amendments to the Material and Methods section as follows below:

QUERY:

How were samples collected?

REPLY:

During September and October 2018, 47 RMBDs were commercially purchased by the investigators in pet shops in six cities within a radius of 300 km of the laboratory, or via internet shops. Four further samples were obtained from a small RMBD producing enterprise which was officially certified based on Hazard Analysis and Critical Control Points (HACCP) hygiene standards through the county veterinary office.

Samples were transported in bags containing cold chain coolants and stored at -20°C. Before analysis, the samples were defrosted 4°C.

QUERY:

Were all samples available in the store purchased? If not, how were they selected?

REPLY:

Only samples that contained uncooked meat or organs that had not undergone any treatment such as pasteurization or drying were purchased. Only RMBDs intended for dogs were included.

QUERY:

Was sampling done in one city or more (and where)?

REPLY:

Sampling was done in six cities within a radius of 300 km of the laboratory.

QUERY:

Were they all different products? Did they all contain meat of Swiss origin? If not, what were the sources?

REPLY:

Thirty-one of the samples contained meat originating from Switzerland, and 20 samples also contained meat imported from Germany.

QUERY:

Were any high pressure pasteurized or otherwise treated to reduce pathogens?

REPLY:

No. This information has been added to the manuscript, please see below.

QUERY:

Were all tested before any provided 'best before' or 'use by' dates?

REPLY:

Yes. For all samples, analysis was performed before the provided 'use by' date.

We have added these details to the manuscript by modifying the Material and Method section:

Line 129: During September and October 2018, 47 RMBDs were commercially purchased by the investigators in pet shops in six cities within a radius of 300 km of the laboratory, or via internet shops. Four further samples were obtained from a small Swiss RMBD producing

enterprise which was officially certified based on Hazard Analysis and Critical Control Points (HACCP) hygiene standards through the responsible county veterinary office. Only samples that contained uncooked meat or organs that had not undergone any treatment such as pasteurization or drying were purchased. Only RMBDs intended for dogs were included.

Samples were transported in bags containing cold chain coolants and stored at -20°C. Before analysis, the samples were defrosted 4°C. For all samples, analysis was performed before the provided 'use by' date.

Line 144: Thirty-one of the samples contained meat originating from Switzerland, and 20 samples also contained meat imported from Germany.

QUERY:

How many colonies were tested per plate? How were they selected?

REPLY:

Every isolate growing on R*Sal* plates was considered *Salmonella* and subjected serotyping. Every potential ESBL producing isolate was picked, subcultured on Brilliance ESBL agar plates and screened for the presence of *bla*_{ESBL} genes by polymerase chain reaction PCR, and identified using MALCI-TOF-MS. The same procedure was applied for colonies growing on plates containing colistin and amikacin.

To further clarify the procedure, we have amended the manuscript as follows:

Line 165: All typical colonies...

Line 174: All colonies with different chromaticity and morphology were subcultured on Brilliance ESBL agar plates for 24 h at 37°C to obtain pure cultures.

Line 181: Colonies were subcultured on selective medium containing 4 mg/L colistin and each pure culture was identified using MALDI-TOF-MS.

Line 188: each pure culture was identified using MALDI-TOF-MS.

Comment #3

Line 235: Standard deviation is not the appropriate measure to report with median.

REPLY:

Line 244 and line 247: standard deviations have been deleted.

Comment #4

Line 236/239: The EU regulation limits are probably best first reported in the methods.

REPLY:

As suggested, the EU regulations are now first mentioned in the Material and Method section:

Line 156: Samples were rated based on the EU hygiene criteria that limit Enterobacteriaceae to 5×10^3 cfu/g for raw meat intended for pet food production.

Comment #5

Was consideration given to any basic statistical comparisons, such as the prevalence between animal sources?

REPLY:

This study was not actually designed to include statistical analysis. Analysis was considered, but the relatively low sample size was too small to support statistical testing.

Reviewer: 2

Comments to the Author(s)

This new version has very little modification compared to the first one. I still think that this is a solid and nice descriptive work that would be more suitable as a short communication in a journal that is more veterinary oriented (Vet mic for example).

Minor comment:

Line 272: formerly instead of formally

REPLY:

Thank you for your comment.

Line 281: The correction has been made.

Reviewer: 3

Comments to the Author(s)

The manuscript was improved and corrections/suggestions were performed as requested. My final comment is that limitations related to ESBL plasmid characterization should be included, since this is also an important public health issue.

REPLY:

Thank you for your comment. We agree that the absence of plasmid analysis is a limitation of this study. To address this issue, we have made an amendment to the manuscript:

Line 388: Notably, plasmid analysis including transfer assays and molecular plasmid typing was not performed in this study, which limits our ability to fully delineate the epidemiologic features of the ESBL-producing Enterobacteriaceae identified in RMBDs.